# Robustness and Tracking Performance Evaluation of PID Motion Control of 7 DoF Anthropomorphic Exoskeleton Robot Assisted Upper Limb Rehabilitation

**DOI:** 10.3390/s22103747

**Published:** 2022-05-14

**Authors:** Tanvir Ahmed, Md Rasedul Islam, Brahim Brahmi, Mohammad Habibur Rahman

**Affiliations:** 1BioRobotics Laboratory, Mechanical/Biomedical Engineering Department, University of Wisconsin Milwaukee, Milwaukee, WI 53211, USA; tanvir@uwm.edu; 2Richard J. Resch School of Engineering, University of Wisconsin-Green Bay, Green Bay, WI 54311, USA; islamm@uwgb.edu; 3Electrical Engineering Department, College Ahuntsic, Montreal, QC H2M 1Y8, Canada; brahim.brahmi@collegeahuntsic.qc.ca

**Keywords:** exoskeleton robot, upper limb rehabilitation, passive rehabilitation exercise, PID motion control, trajectory tracking

## Abstract

Upper limb dysfunctions (ULD) are common following a stroke. Annually, more than 15 million people suffer a stroke worldwide. We have developed a 7 degrees of freedom (DoF) exoskeleton robot named the smart robotic exoskeleton (*SREx*) to provide upper limb rehabilitation therapy. The robot is designed for adults and has an extended range of motion compared to our previously designed ETS-MARSE robot. While providing rehabilitation therapy, the exoskeleton robot is always subject to random disturbance. Moreover, these types of robots manage various patients and different degrees of impairment, which are quite impossible to model and incorporate into the robot dynamics. We hypothesize that a model-independent controller, such as a PID controller, is most suitable for maneuvering a therapeutic exoskeleton robot to provide rehabilitation therapy. This research implemented a model-free proportional–integral–derivative (PID) controller to maneuver a complex 7 DoF anthropomorphic exoskeleton robot (i.e., *SREx*) to provide a wide variety of upper limb exercises to the different subjects. The robustness and trajectory tracking performance of the PID controller was evaluated with experiments. The results show that a PID controller can effectively control a highly nonlinear and complex exoskeleton-type robot.

## 1. Introduction

Rehabilitation robots can significantly reduce the burden of therapists by providing repetitive and precise therapy to people with upper limb impairment for a long duration of time. With the increasing number of stroke occurrences, continuously more patients incur upper limb mobility impairment [1,2,3,4,5]. In addition, upper limb dysfunctions (ULD) often originate from spinal cord injuries, sports injuries, trauma or occupational injuries. Returning lost mobility in patients with ULD is imperative as the human upper limb involves conducting various daily activities. Many research prototypes of upper limb rehabilitation robots have been developed to date, and they can be categorized into two types: The end-effector type and the exoskeleton type. The end-effector-type robots are lightweight, small in structure, and most suitable for end-point exercise only (such as pulling/pushing or following a path). Still, this type of robot cannot provide joint-based exercises (such as elbow/shoulder joint flexion/extension). On the other hand, exoskeleton-type robots are relatively heavy and robust, but they can control individual upper limb joints and mimic every variety of recommended upper limb exercises [6,7,8,9,10].

In early research on rehabilitation robots, the researcher proposed end-effector-based robots to provide rehabilitation therapy in planar arm motion [11,12,13,14,15,16]. The prime limitation of these robots is the inability to provide controlled movement to individual joints of the upper limb. Even though the robot workspace is larger, this constraint significantly decreases the rehabilitation workspace. Many research groups have developed exoskeleton-type robots that can control each joint individually to address this barrier [6,8,9,10,17,18,19,20,21,22,23,24,25,26,27,28]. Although several control algorithms/methods are developed to meet rehabilitation robot control requirements, the performance of human–robot interaction is still very insufficient.

Exoskeleton-type robots for providing rehabilitation therapy are constantly subject to unpredictable disturbances from the patient while providing therapy. The controller must adapt to these disturbances stemming from the patient, system, and environment. In addition, the exoskeleton robot interacts with a wide variety of patients with varying degrees of impairment, which are almost hard to model and incorporate into the robot dynamics due to the time-changing nature of the human–robot interaction. Most of the existing exoskeleton robot research has designed their control approach based on the simplified robot model [7,29,30,31,32,33]. Therefore, their performance is not very effective when it comes to managing unpredictable disturbances.

Moreover, an inaccurate/simplified robot model is often considered one of the major causes of the poor performance of the nonlinear control. Some research groups used a model-free sliding mode control to manage uncertainty [20,29,34,35]. However, in these cases, control input data have been chattered by a combination of uncertainties. To overcome this issue, some researchers proposed hybrid control approaches (e.g., fuzzy-sliding mode, fuzzy-neuro adaption, sliding mode with an artificial neural network, etc.) [25,33,36,37]. The main drawback of these control approaches is that they require heavy computation.

Motivated by the discussions mentioned above, in this paper, we hypothesize that a model-independent controller, such as a PID controller, is most suitable for maneuvering an exoskeleton-type robot to provide rehabilitation therapy. To test this hypothesis in this research, we tested the developed *SREx* robot with a model-free PID controller to provide every variety of upper limb exercises to the five healthy subjects. In typical rehabilitation settings, upper limb movement therapies are provided within the velocity range of 15 to 53°/s [38,39], depending on the patients’ ULD. Furthermore, to test the robustness and tracking performance of the controller, the *SREx* was tested with (a) individual and multi-joint movement trajectories with a varying velocity ranging from 15 to 72°/s; (b) continuous resistance force while providing therapy as a mimic of providing therapy to the individuals with upper limb spasticity; and (c) sudden jerk/involuntary movements that are often observed among the patients with a stroke. Experiment results guarantee that a model-free PID controller can be most suitable for maneuvering a therapeutic exoskeleton robot to provide rehabilitation therapy. It is worth mentioning that several researchers have carried out simulation studies and experiments with the PID controller for maneuvering exoskeleton robots for rehabilitation of human upper or lower limbs [40,41,42,43]. However, to the authors’ knowledge, none of the prior work thoroughly investigated the effectiveness of a well-established conventional PID controller to provide robot-aided rehabilitation to human subjects’ varying conditions from the controller performance perspective.

The paper’s main contribution is two-fold, introducing a new therapeutic robot, *SREx*, and its control implementation with a model-independent (i.e., conventional PID) controller to provide every variety of upper limb rehabilitation exercises. These exercises are designed to emulate varying conditions of stroke survivors’ upper limbs. As healthy human subjects have been used during the experiments rather than actual patients, the different interactions between humans and robot are presented in this paper. These interactions during the experiments are reflected by time-series data collected from the force/torque sensor installed on the robot. The remainder of the article is organized as follows: Section 2 presents a brief discussion regarding *SREx*, Section 3 discusses the kinematics of the *SREx*, Section 4 depicts the fundamentals of the PID control approach, Section 5 demonstrates the experimental setup of the overall research and illustrates the result of this study, and finally, Section 6 draws the conclusions of the study.

## 2. Smart Robotic Exoskeleton (*SREx*)

The smart robotic exoskeleton (*SREx*) presented in this research is the second prototype of our previously developed ETS-MARSE exoskeleton robot [24]. The *SREx* was designed based on the upper-limb biomechanics with an extended range of motion. It is a 7 degrees of freedom (DoF) exoskeleton robot (see Figure 1a), comprised of a 3 DoF shoulder motion support part, a 2 DoF elbow and forearm motion support, and a 2 DoF wrist motion support part. The exoskeleton was designed for use by adults. Provisions were included in the design to adjust the link length based on the user’s needs. The robot is designed to be worn on the lateral side of the right upper limb [44]. Its shoulder motion support component provides shoulder joint vertical and horizontal flexion/extension and internal/external rotation. The elbow and forearm motion support components provide joint elbow flexion/extension and forearm pronation/supination, and the wrist joint support component provides wrist joint radial/ulnar deviation and flexion/extension. Figure 2 illustrates the workspace of the *SREx*. The entire exoskeleton arm was fabricated primarily in aluminum. Brushless DC motors (Maxon EC 90 and 45 motors) integrated with harmonic drives were used to actuate the developed *SREx*. A 6-axis force/torque (F/T) sensor is instrumented underneath the wrist handle to measure the instantaneous human–robot interaction forces. Table 1 and Table 2 summarize the technical specification and mass/inertia properties of the *SREx*, respectively.

***Safety Consideration****:* Provision has been included in the design of *SREx* to add adjustable mechanical stoppers at each joint to limit the joints’ rotation of *SREx* corresponding to the users’ range of motion to ensure the safety of the robot users. A switch is also placed to disable the control signal in an emergency. If the emergency switch is activated during the exercises, the robot is programmed to remain in its current location. This accommodation ensures that the human user stays safe during the crisis. In addition to these hardware safety features, software safety features were added to the control algorithm, such as limiting the movement ranges of the joints based on the patient’s needs, limiting the speed of the joints, limiting the joint torques, and limiting the voltage delivered to the motor drivers.

## 3. Kinematics of *SREx*

The kinematic model of the *SREx* was developed based on modified Denavit–Hartenberg (DH) notations. To obtain the DH parameters, we assume that the origin of the coordinate frames (i.e., the link-frames that map between the successive axes of rotation) coincide with the joint axes of rotation and have the same order, i.e., frame {1} coincides with Joint 1, frame {2} with Joint 2, etc. As shown in Figure 1b, the joint axes of rotation of the *SREx* corresponding to the human upper limb are indicated by red arrowheads (i.e., *Z_i_*). In this model, Joints 1, 2, and 3 together constitute the shoulder joint, where Joint 1 corresponds to horizontal flexion/extension, Joint 2 represents vertical flexion/extension, and Joint 3 corresponds to internal/external rotation of the shoulder joint. Joint 4 represents the elbow joint and is located at a distance *d_e_* (length of the upper arm) from the shoulder joint. Joint 5 corresponds to forearm pronation/supination. As shown in Figure 1b, Joints 6 and 7 intersect at the wrist joint, at a distance *d_w_* (length of the forearm) from the elbow joint. Joint 6 is responsible for wrist joint radial/ulnar deviation, and Joint 7 provides the wrist joint flexion/extension motion. The F/T sensor is located at a distance *d_T_* from the wrist joint. The modified DH parameters corresponding to the link frames assignment shown in Figure 1b are summarized in Table 3. These DH parameters are used to achieve the homogeneous transfer matrix representing the positions and orientations of a reference frame with respect to the fixed reference frame.

**Assumption** **1.**
*The fixed reference frame*

{0}

*is located at a distance ds apart from the first reference frame*

{1}

*of SREx.*


**Assumption** **2.**
*Frames {1} and {2} are located at the same point at the shoulder joint.*


**Assumption** **3.**
*Frames {4} and {5} are at the same point at the elbow joint.*


**Assumption** **4.**
*Frames {5}, {6}, and {7} are located at the same point at the wrist joint.*


The general form of homogeneous transformation that relates frame {i} relative to the frame {i−1} [45] can be expressed by Equation (1).
(1)Tii−1=[Rii−13×3Pii−13×101×31]
where Rii−1 is the rotation matrix that describes the frame {i} relative to frame {i−1} and can be expressed as:(2)Rii−1=[cosθi−sinθi0sinθicosαi−1cosθicosαi−1−sinαi−1sinθisinαi−1cosθisinαi−1cosαi−1]
and Pii−1 is the vector that locates the origin of the frame {i} relative to frame {i−1} and can be expressed as:(3)Pii−1=[ai−1−sαi−1dicαi−1di]T

Using Equations (1)–(3), the homogenous transformation matrix that relates frame {7} to frame {0} can be obtained by multiplying individual transformation matrices (Tii−1).
(4)T70=[T10·T21T43·T54·T65·T76]

The single transformation matrix thus found from Equation (4) represents the positions and orientations of the reference frame {7} attached to the wrist joint (axis 7) with respect to the fixed reference frame {0} attached to the *SREx*’s base.

## 4. Control

This research uses a model-free decoupled PID control to maneuver the *SREx* to provide upper limb rehabilitation therapy. PID control is the most widely used control technique in industrial applications [45]. Moreover, classical PID controller’s stability is ensured with the second order nonlinear systems, such as *SREx*, specifically for the time-independent desired joint position. Since the desired position is time-varying in this paper’s context, it may not be arbitrary, but should have a very particular form. In general, it is impossible to obtain an expression in closed form for the time-varying of the desired trajectory, thus the closed-loop must be solved numerically. However, the controller stability can be ensured by leveraging the gain tuning procedure described by Kelly et al. [46].

Furthermore, PID is simple in design and efficient in computation. Moreover, it is considered a robust control technique. The general layout of the PID control approach is depicted in Figure 3. The joint torque commands (PID controller output) of the *SREx* can be expressed by Equation (5):(5)τ=KP(θd−θ)+KV(θ˙d−θ˙)+KI∫(θd−θ)dt
where θd,θ∈ℝ7 are the vectors of desired and measured joint angles, respectively, θd˙,θ˙∈ℝ7 are the vectors of desired and measured joint velocities, respectively, ***K_P_*****, *K_V_*, *K_I_*** are the diagonal positive definite gain matrices, and τ∈ℝ7 is the joint torque vector. Let the error vector ***E*** and its derivative be:(6)E=θd−θ; E˙=θd˙−θ˙

Therefore, Equation (5) can be re-formulated as an error equation:(7)$$\tau =K_{P}E+K_{V}\dot{E}+K_{I}\int Edt$$

The Equation (7) is decoupled, therefore, the individual torque command for each joint would be as follows:(8)$$\tau_{i}=K_{P_{i}}e_{i}+K_{V_{i}}\dot{e_{i}}+K_{I_{i}}\int e_{i}dt$$
where i=1, 2,⋯7 represents the robot’s seven joints, $\theta_{i},\;\theta_{d_{i}\;}$ are measured and desired joint angles, respectively, ${\dot{\theta }}_{i},\;{\dot{\theta }}_{d_{i}\;}$ are measured and desired joint velocities, respectively, $e_{i}=\theta_{d_{i}\;}-\theta_{i},$ ${\dot{e}}_{i}={\dot{\theta }}_{d_{i}\;}-{\dot{\theta }}_{i}$ are position tracking errors and velocity error for joint ***i***, respectively, and
$$E={\left[\begin{array}{cccc} e_{1} & e_{2} & \cdots & e_{7} \end{array}\right]}^{T},$$
$$\dot{E}={\left[\begin{array}{cccc} \dot{e_{1}} & \dot{e_{2}} & \cdots & \dot{e_{7}} \end{array}\right]}^{T},\;$$
$$K_{P}=diag{\left[\begin{array}{cccc} K_{P_{1}} & K_{P_{2}} & \cdots & K_{P_{7}} \end{array}\right]}^{T}\;\;$$
$$K_{V}=diag{\left[\begin{array}{cccc} K_{V_{1}} & K_{V_{2}} & \cdots & K_{V_{7}} \end{array}\right]}^{T},\;$$
$$\;K_{I}=diag{\left[\begin{array}{cccc} K_{I_{1}} & K_{I_{2}} & \cdots & K_{I_{7}} \end{array}\right]}^{T}\;$$

## 5. Experiments and Results

### 5.1. Experimental Setup

The experimental setup and the control architecture of the *SREx* system are depicted in Figure 4 and Figure 5, respectively. Hall sensors embedded with each joint motor of the *SREx* measure the joint angle and are sampled at 100 µs and then filtered with a second order filter (see control architecture in Figure 5) with a damping factor ζ = 0.90 and natural frequency ω_0_ = 30 rad/s before being sent to the PID controller. The rationale for selecting the second order filter for the *SREx* system is that this filter is stable and easy to implement. In addition, its time of response is manageable only by tuning gain, damping, and frequency parameters. At the start of the experiments, the gains of PI controller of the current loop have been tuned experimentally to achieve optimum response.

As shown in Figure 5, the joint torque commands from the output of the PID controller are converted to motor currents, and then to voltage to generate reference values that serve as drive commands for the motor drivers (Brushless PWM Servo Amplifier, Model: ZB12A8, AMC, block diagram can be found in Appendix A). Note that the controller (PID) updates the torque commands every 500 µs and is executed in RT OS (left dotted box, Figure 5). Furthermore, to realize the real-time control of the *SREx* and to ensure that the proper current commands are sent to the driver, a PI controller (right dotted box, Figure 5) was added to minimize the differences between the desired and measured currents (i.e., the error command to PI controller). The PI controller runs ten times faster than the torque control loop and is executed in a field-programmable gate array (FPGA). The current signals measured from the current monitor output of motor drivers are sampled at 50 µs and are then filtered with a second order filter (see Figure 5) with a damping factor ζ = 0.90 and natural frequency ω_0_ = 1000 rad/s prior to being sent to the PI controller. In this control architecture, the high-level control in RT-PC and low-level control in FPGA were implemented. In this case, the low-level control loop (inner loop) needs to be faster than the high-level control loop (outer loop). The secondary process (low-level control) must react to the secondary controller’s efforts at least three or four times faster than the primary process (high-level control) reaction to the primary controller. This approach allows the secondary controller (low-level control) enough time to compensate for inner loop disturbances before they can adversely affect the primary process. Note that the controller’s (PID) gains used for the experiments were found by trial and error after establishing a basis for gain tuning using the method described by Kelly et al. [46] and are as follows:$$K_{P}=diag\left[\begin{array}{ccccccc} 300 & 750 & 450 & 700 & 300 & 420 & 650 \end{array}\right],$$
$$K_{V}=diag\left[\begin{array}{ccccccc} 40 & 250 & 120 & 200 & 100 & 280 & 400 \end{array}\right],\;$$
and
$$K_{I}=diag\left[\begin{array}{ccccccc} 30 & 35 & 15 & 25 & 10 & 6 & 1.5 \end{array}\right].$$

### 5.2. Experiments and Results Analysis

Experiments were conducted with healthy male human subjects (age: 25–44 years; height: 161.5–177.8 cm; weight: 63–82 kg; the number of subjects: 5) seated on a chair wearing the *SREx* on the lateral side of the right hand, as shown in Figure 1a. Therefore, the *SREx* was subjected to continuous, highly coupled nonlinear disturbance during the exercise period. The goal of the experiments is to evaluate the PID controller’s performance (a) to maneuver this complex *SREx* robot to provide passive arm movement therapy to individual and multi-joint upper limb motion, and (b) to verify the robustness and trajectory tracking performance of the controller. It would be pertinent to mention that in a typical rehabilitation setting, upper limb movement therapies are provided at a speed of 15 to 53°/s [38,39], depending on the patients’ ULD. To evaluate the performance of the developed *SREx* system and to test this hypothesis in this research, the developed *SREx* robot was maneuvered with a model-free PID controller to provide every variety of upper limb exercises. Furthermore, to test the robustness and tracking performance of the controller, the *SREx* was tested with (a) different subjects (see subject parameters in Table 4) for individual and multi-joint movement trajectories with a varying velocity ranging from 15 to 72°/s; (b) continuous resistance force while providing therapy as a mimic of providing therapy to an individual with upper limb spasticity; and (c) sudden jerk/perturbation that is often observed among the patients with a stroke.

#### 5.2.1. Individual Joint Movement Passive Exercise

We conducted individual upper limb joint movement exercises, such as elbow flexion/extension, forearm pronation/supination, and shoulder vertical flexion/extension (F/E) motion exercises with Subjects A, B, and C. In this paper, we have included results for elbow flexion/extension only as both shoulder vertical F/E and elbow F/E are against the gravity.

*Elbow Flexion/Extension Exercise:* Figure 6 shows the experimental results of elbow motion, where the *SREx* repeatedly provides elbow flexion/extension motion to Subject A from 10 to 130° at a varying velocity from 15 to ~72°/s. The first row in the figure compares the desired joint angles (i.e., reference trajectories/exercise, dotted line) to measured joint angles (i.e., measured trajectories, solid line). The second row shows the tracking error; the third row compares the desired velocity (dotted line) against the measured velocity (solid line); the fourth row shows the joint torques. The fifth and sixth rows display human–robot interaction forces and torques, respectively. As seen in Figure 6, the exercise was repeated in three cycles and at different velocity ranges to show *SREx*’s performance in providing passive therapies to subjects. The first cycle represents the slow velocity range (trajectory time: ~2 to ~16 s), the second cycle represents the medium velocity range (trajectory time: ~17 to ~26 s), and the third cycle represents the fast velocity range (trajectory time: ~27 to ~32 s). The peak velocity of the first cycle was observed around ~25.6°/s, and for the second and third cycles the peak velocity was observed around 40 and 71.9°/s, respectively. Throughout this paper, the different velocities were predefined based on current practices and the need for rehabilitation therapies [38,39]. Note that to check/validate the robustness of the controller, the PID gains used in this research were tuned only for the medium velocity range (~35°/s). We hypothesized that the same PID control gains are adequate to maneuver the complex *SREx* robot to provide rehabilitation therapy in all possible rehabilitation scenarios, including exercises performed at different velocities with different subjects.

In Figure 6, the left column presents the results of Case 1, where the subject remained passive (i.e., no voluntary movement by the subject). On the other hand, the middle column illustrates the results of Case 2, where the subject was asked to resist the motion (see force/torque plots, Figure 6, middle column, fifth and sixth rows) to simulate the case of spasticity. The maximum resistive force observed in this case was around 21.03 N, detected from the F/T sensors at the wrist joint. The right column presents the results of Case 3, where the subject was asked to exert sudden jerk/perturbation to simulate the case of involuntary movement (see force/torque plots, Figure 6, right column, fifth and sixth rows) that is often observed among the patients with a stroke. The maximum jerk force observed in this case was around 12.27 N, detected from the F/T sensors at the wrist joint. It can be seen from the tracking plots (first row, Figure 6), that the measured and desired trajectories are entirely overlapped in all three cases despite resistance to motion and sudden jerk. Therefore, the tracking errors are relatively minor (see tracking error plots, second row, Figure 6), which is less than 1° in the first and second cycles and less than 2.5° in the third cycle (in the case of high speed). By comparing the force/torque sensor data trails for three cases, one can recognize the differences in human–robot interaction, which stems from emulated conditions of the same subject.

*Forearm Pronation/Supination Exercise:*Figure 7 shows the experimental results of forearm motion, where the *SREx* repeatedly provides pronation/supination motion (see schematic in Figure 2e) to Subject C, ranging from −70 to 70°. As seen in Figure 7, the exercise was repeated in three cycles and performed at varying velocities. The peak velocity of the first cycle (slow velocity cycle, trajectory time: ~0.5 to ~12.5 s) was observed around ~26.2°/s, and for the second (medium velocity cycle, trajectory time: ~22.5 to ~12.5 s) and third cycles (fast velocity cycle, trajectory time: ~29.5 to ~32 s) were observed around 42 and 59.9°/s, respectively.

In Figure 7, the left column presents the results of Case 1, where the subject remained passive (i.e., no voluntary movement by the subject). The middle column illustrates the results of Case 2, where the subject was asked to resist the motion (see force/torque plots, Figure 7, middle column, fifth and sixth rows) to simulate the case of spasticity. The maximum resistive force observed in this case was around 12.19 N. Finally, the right column presents the results of Case 3, where the subject was asked to exert sudden jerk/perturbation to simulate the case of involuntary movement (see force/torque plots, Figure 7, fifth and sixth rows) that is often observed among the patients with a stroke. The maximum jerk force observed in this case was around 8.13 N. It can be seen from the tracking plots (first row, Figure 7) that the controller performance was excellent despite providing the resistance to motion, and the measured trajectory completely overlaps with the desired trajectory, similar to the previous exercises. The tracking error was also observed to be relatively small in this exercise (less than 1° in the first two cycles and less than 2.5° in the third cycle). Note that although in rehabilitation robotics the trajectory tracking error is not a vital matrix to measure control performance, but rather the chattering-free trajectory tracking and robustness of the controller matter, the experimental results warrant the PID controller’s excellent performance in trajectory tracking and robustness.

#### 5.2.2. Composite Passive Joint Movement Exercises (Diagonal Reaching Motion)

To further evaluate the robustness and trajectory tracking performance of the PID control, a multi-joint movement exercise involving the simultaneous motion of the shoulder and elbow joint, representing diagonal reaching movement exercise, was performed with the three subjects (Subjects A, B, and C).

Figure 8 shows the Cartesian trajectory tracking plots of diagonal reaching motion (i.e., robot/subject’s end-effector position plots) in three cases when Subject B (a) remained passive, (b) resisted the motion, and (c) exerted sudden jerk motion. It can be seen from the plots that the exercises were performed at three different velocity ranges (first cycle: Slow velocity, trajectory time: ~0.5 to ~6.5 s; second cycle: Medium velocity, trajectory time: ~7 to ~11 s; third cycle: Fast velocity, trajectory time: ~11.5 to ~13.5 s). In addition, tracking errors vary with the increase of velocity, but they are minimal.

Figure 8a shows the experimental results of Cartesian trajectory tracking plots of diagonal reaching motion in Subject B, where the subject remained passive (i.e., no voluntary movement by the subject); the corresponding shoulder (Joints 1 and 2) and elbow joint (Joint 4) motions are plotted in Figure 9a. As seen in Figure 8a and Figure 9a, the exercise was repeated in three cycles and performed at varying velocities, ranging from 15 to 60°/s. The peak velocity of the first cycle was observed around ~20°/s, and for the second and third cycles the peak velocity was observed at 30 and 60°/s, respectively. It can be seen from Figure 8a and Figure 9a, that both Cartesian and joint based tracking errors are minimal, where the Cartesian tracking error is observed <0.9 cm in first cycle, <1.5 cm in second cycle, and <2 cm in third cycle, whereas the joint based tracking error is found <0.5° in first and second cycles, <1.5° in third cycle.

Figure 8b illustrates the Cartesian trajectory tracking plots of a similar diagonal reaching motion as in Figure 8a, but the subject was asked to resist the motion to simulate the case of spasticity. The corresponding joint movements are plotted in Figure 9b. It can be seen from Figure 8b and Figure 9b that both Cartesian and joint based tracking errors are minimal, where the Cartesian tracking error is observed <1 cm in first cycle, <2 cm in second cycle, and <2.6 cm in third cycle, whereas the joint based tracking error is found <1.5° in first and second cycles, <3° in third cycle. Figure 8c demonstrates the Cartesian trajectory tracking plots of a similar diagonal reaching motion as Figure 8a. However, the subject was asked to exert sudden jerk/perturbation to simulate the case of involuntary movement that is often observed among the patients with a stroke. The corresponding shoulder and elbow joint movements are plotted in Figure 9b. As seen in Figure 8c and Figure 9c, similar to the previous two trials, the exercise was repeated in three cycles and performed at a varying velocity, ranging from 15 to 60°/s. Furthermore, both Cartesian and joint based tracking errors are minimal, where the Cartesian tracking error is observed <1 cm in first cycle, <2 cm in second cycle, and <3.8 cm in third cycle, whereas the joint based tracking error is found <1.5° in first and second cycles, <4° in third cycle. From the force/torque plots vs. error plots in Figure 10, it is evident that errors slightly increase when subjects resist the motion and/or add sudden jerk motion.

Figure 10 presents human–robot interaction force/torque plots corresponding to the diagonal reaching exercise for the three different cases, where Figure 10a corresponds to the first case when the subject remained passive during the motion, indicating that the subject provided no voluntary movement during the motion provided by the robot. Figure 10b represents the second case when the subject resisted the motion during the motion. Figure 10c presents the third case when the subject provided perturbation or sudden jerk force during the motion. The first row presents the joint position tracking error for all three joints. The second and third rows show the force and torque values captured through the force/toque sensor during three different cases. Figure 10a shows that the tracking error of Joint 1 was at <2°, where the maximum interaction force at the wrist handle was observed at around 3.04 N, even when the subject remained passive (first case). For the second case, the maximum resistance to motion force was observed to be around 18.45 N (see Figure 10b, second row), which is detected from the F/T sensor. It can be seen from the second row in Figure 10c that the maximum jerk force is around 12.18 N for the third case. The critical difference between the second and third case is the nature and magnitude of force and torque exerted by the subject, which can be clearly seen in Figure 10a–c (second and third rows). The associated tracking error plots provide insight into the correlation between the exerted force by the Subject and *SREx*’s capability to follow the predefined passive therapy.

#### 5.2.3. All Joints Simultaneous Motion (Passive)

To further evaluate the performance of the *PID controller* regarding dynamic trajectory tracking and robustness, another experiment involving simultaneous movements of all upper limb joints, i.e., shoulder, elbow, forearm, and wrist joint movements (7DoF), was performed with Subjects D and E. In this paper, we have presented the results for Subject D.

Figure 11 shows the experimental results, where *SREx* provides simultaneous motion of all upper limb joints to Subject D. The exercise was repeated for two cycles and performed at varying velocity ranges in each cycle (slow velocity cycle, trajectory time: ~1 to ~17 s; fast velocity cycle, trajectory time: ~18 to ~24 s), ranging from 15 to 30°/s. During this trial, the subject remained passive (i.e., no voluntary movement by the subject). On the other hand, Figure 12a illustrates the Cartesian trajectory tracking plots of the exercise (maximum tracking error observed is less than 1 cm). It can be seen from the tracking error plots in Figure 11 that the controller performance was excellent, as, again, the tracking error was relatively small (less than 1.5°), except for Joint 3. The maximum tracking error was observed in Joint 3, around 2.5°, which is also minimal.

To further substantiate the performance of the PID controller, a similar exercise to Figure 11 and Figure 12a was performed with *SREx*, where the same subject was asked to resist the motion to simulate the case of spasticity/disturbances. Figure 12b illustrates the Cartesian trajectory tracking plots of the exercise, where the Cartesian tracking error was observed as less than 1.5 cm. It can be seen from the tracking error plots in Figure 12b that the controller performance was excellent, as, again, the tracking error was relatively small (less than 1.5°), except for Joint 3. The maximum tracking error was observed in Joint 3, around 2.5°, which is also minimal.

Finally, Figure 13 presents human–robot interaction, force/torque plots, and associated position tracking errors for individual robot joints in both cases (see Figure 13a,b). The tracking error and force/torque trails show that when the subject provides perturbation during the robot’s predefined motion, the joint tracking errors increase but remain minimal and stable throughout this complex motion. When the subject remained passive, the maximum interaction force was observed around 3 N (see Figure 11, Figure 12a and Figure 13a), detected from the F/T sensor at the wrist handle. For the case where the subject resisted the robot’s motion, the maximum interaction/resistive force is observed around 40 N (Figure 12b and Figure 13b).

As a result, the experimental results presented in Figure 6, Figure 7, Figure 8, Figure 9, Figure 10, Figure 11, Figure 12 and Figure 13 guarantee that a model-free PID controller can effectively maneuver the developed *SREx* to provide individual and multi-joint movement upper limb exercises at a varying velocity and different exercise scenarios, including resistance to motion exercise and sudden involuntary movement exercise. Overall, the PID controller is found to be very robust in all cases.

#### 5.2.4. Statistical Comparison

Table 5 summarizes and compares the experimental results of Subjects A, B, and C of elbow joint flexion/extension motion. Note that the experimental results of only Subject A are presented in Figure 6. Therefore, Table 5 compares the root mean square (RMS) error and peak tracking error of elbow joint flexion/extension motion of Subjects A, B, and C in different experiment conditions, including when the subject (a) remained passive, (b) resisted the exercise, and (c) exerted sudden jerk motion. It can be seen from Table 5 that for this particular exercise, the maximum RMS error is observed in the case of ‘*resistance to motion exercise*’, which is 0.64°, where Subject A’s resistive force was found around 24.38 N. The maximum peak tracking error, which is 2.06° was also observed in the case of ‘*resistance to motion exercise*’, but for Subject B. The exact magnitude of peak tracking error (2.06°) was also observed in the case of the ‘*sudden jerk*’ event of Subject A’s flexion/extension motion, where Subject A’s jerk force was found around 12.27 N. We can see that as the resistive forces increase, the tracking errors also increase.

Furthermore, it is observed from the data presented in Table 5 that during ‘*resistance to motion exercise*’, Subject A exerted approximately six times the forces observed in the case of ‘passive’ motion; however, the RMS error was increased by 0.22°, and the peak tracking error was raised by 0.45°, only. Similarly, during ‘*sudden jerk motion*’, Subject A exerted approximately four times the forces observed in the case of ‘passive’ motion; however, the RMS error was increased by 0.13°, and the peak tracking error was raised by 0.57°, only. On the other hand, during the same ‘*resistance to motion exercise*’, Subject B exerted approximately 3.6 times the forces observed in the case of ‘*passive*’ motion; however, the RMS error was increased by 0.13°, and the peak tracking error was raised by 0.57°, only. Similarly, during ‘*sudden jerk motion*’, Subject B exerted approximately 2.5 times the forces observed in the case of ‘*passive*’ motion; however, the RMS error was increased by 0.03°, and the peak tracking error was raised by 0.11°, only. Subject C’s data also show a similar pattern, where the RMS and peak tracking error increased slightly with the resistive/jerk force.

Table 6 summarizes and compares the experimental results of Subjects A, B, and C of forearm pronation/supination motion. Note that the experimental results of only Subject C are presented in Figure 7. Therefore, Table 6 compares the root mean square (RMS) error and peak tracking error of forearm pronation/supination motion of Subjects A, B, and C in different experiment conditions, including when the subject (a) remained passive, (b) resisted the exercise, and (c) exerted sudden jerk motion. It can be seen from Table 6 that for this particular exercise, the maximum RMS error is observed in the case of ‘*resistance to motion exercise*’, which is 1.07°, where Subject B’s resistive force was found around 19.55 N. The maximum peak tracking error, which is 2.69°, was also observed in the case of ‘*resistance to motion exercise*’.

Furthermore, it is observed from the data presented in Table 6 that during ‘*resistance to motion exercise*’, Subject B exerted approximately 9.8 times the forces observed in the case of ‘passive’ motion; however, the RMS error was increased by 0.24°, and the peak tracking error was raised by 0.63°, only. Similarly, during ‘*sudden jerk motion*’, Subject B exerted approximately 6.18 times the forces observed in the case of ‘passive’ motion; however, the RMS error was increased by 0.12°, and the peak tracking error was decreased by 0.05°, only. On the other hand, during the same ‘*resistance to motion exercise*’, Subject C exerted approximately 3.61 times the forces observed in the case of ‘*passive*’ motion; however, the RMS error was increased by 0.07°, and the peak tracking error was raised by 0.29°, only. Similarly, during ‘*sudden jerk motion*’, Subject C exerted approximately 2.41 times the forces observed in the case of ‘*passive*’ motion; however, the RMS error remained the same, and the peak tracking error was raised by 0.29°, only. Subject A’s data also show a similar pattern, where the RMS and peak tracking error increased slightly with the resistive/jerk force.

Table 7 summarizes and compares the experimental results of Subjects A, B, and C of diagonal reaching motion. Note that the experimental results of only Subject B are presented in Figure 8, Figure 9 and Figure 10. Therefore, Table 7 compares the root mean square (RMS) error and peak tracking error of shoulder joint flexion/extension motion of Subjects A, B, and C in different experiment conditions, including when the subject (a) remained passive, (b) resisted the exercise, and (c) exerted sudden jerk motion. It can be seen from Table 7 that for this particular exercise, the maximum RMS error is observed in the case of ‘*sudden jerk*’ event, which is 0.65°, where Subject A’s resistive force was found around 11.6 N. The maximum peak tracking error, which is 1.91°, was also observed in the case of ‘*sudden jerk event*’ but for Subject B.

Furthermore, it is observed from the data presented in Table 7 that during ‘*resistance to motion exercise*’, Subject B exerted approximately six times the forces observed in the case of ‘passive’ motion; however, the RMS error was increased by 0.22°, and peak tracking error was raised by 0.45°, only. Similarly, during ‘*sudden jerk motion*’, Subject A exerted approximately 6.1 times the forces observed in the case of ‘passive’ motion; however, the RMS error was increased by 0.13°, and the peak tracking error was raised by 0.57°, only. On the other hand, during the same ‘*resistance to motion exercise*’, Subject B exerted approximately 3.6 times the forces observed in the case of ‘*passive*’ motion; however, the RMS error was increased by 0.11°, and the peak tracking error was raised by 0.38°, only. Similarly, during ‘*sudden jerk motion*’, Subject B exerted approximately four times the forces observed in the case of ‘*passive*’ motion; however, the RMS error was increased by 0.04°, and the peak tracking error was raised by 0.37°, only. The data for Subjects A and C also show a similar pattern, where the RMS and peak tracking error increased slightly with resistive/jerk force.

A minimal RMS and peak tracking error increase is observed with a significant resistive/jerk force. Therefore, we found a co-relation between the RMS tracking error vs. resistive/jerk force, and peak tracking error vs. resistive/jerk force. Therefore, the statistical data presented in Table 5, Table 6 and Table 7 conclude that the PID controller demonstrated an extremely robust performance.

Table 8 summarizes the root mean square (RMS) error of each joint of the *SREx* for the multi-joint movement exercise for Subjects D and E, where the subjects remained passive during the robot’s predefined motion. The results show an excellent tracking performance of the PID controller with the RMS tracking error around or below 0.5° for all the joints except for Joint 3, where the maximum tracking error was observed around 1.4°, which is still minimal.

Table 9 summarizes the root mean square (RMS) error of each joint of the *SREx* for the multi-joint movement, where the subjects were asked to resist the robot’s motion. These results also show excellent tracking performance of the PID controllers, with the RMS tracking error observed around or below 0.7° for all the joints except for Joint 3, where the maximum tracking error was observed at around 1.59°.

It is evident from the results presented in Figure 6, Figure 7, Figure 8, Figure 9, Figure 10, Figure 11, Figure 12 and Figure 13 and from the statistical comparison shown in Table 5, Table 6, Table 7, Table 8 and Table 9 that a PID controller can efficiently drive the developed 7DoF *SREx* to provide single and multi-joint movement upper limb exercises. In the research conducted by Wu et al., the authors presented the control of a 7 DoF exoskeleton-type for providing elbow flexion/extension and forearm pronation/supination exercises with a model-based adaptive sliding mode controller with disturbance observer (ASMCDO) [47]. The exercises were conducted with three subjects while they remained passive throughout the motions. The authors compared the performance of their proposed controller with a conventional terminal sliding mode controller (TSMC), which is another model-based controller. Although their proposed controller showed better tracking performance compared to TSMC, however, even in the passive condition of the subjects’, the ASMCDO produced a maximum absolute error (MAXE) greater than 4° for elbow flexion/extension exercise and more than 4.5° for forearm pronation/supination exercise. The authors also found that for the trials with passive exercise experiments, the root mean square error (RMSE) was greater than 1.7°. Whereas the conventional PID controller in this study produced MAXE or peak error to be less than ~1.5° for elbow flexion/extension exercise and less than ~2° for forearm pronation/supination exercise (see Table 6 and Table 7). In addition, using the conventional PID controller, the RMSE was found to be less than 0.5° and less than 0.95° for elbow and forearm exercise, respectively.

Moreover, the PID controller is found to be robust in providing therapy to the different subjects at varying velocities and different passive exercise scenarios, including the subject’s relaxed state during exercise, resistance to motion exercise, and sudden involuntary movement during exercise. Furthermore, the robustness and stability of the PID controller are confirmed in the experimental results in Figure 6, Figure 7, Figure 8, Figure 9, Figure 10, Figure 11, Figure 12 and Figure 13, where the participants’ artificially induced resistance to motion and imitation of spasticity are often observed in patients with stroke. The statistical data presented in Table 5, Table 6, Table 7, Table 8 and Table 9 also conclude that the PID controller demonstrated extremely robust performance. With a substantial resistance/jerk force, a slight increase in RMS, and an observed peak tracking error. Overall, the PID controller is found to be very robust in all cases.

## 6. Conclusions

This study introduces the smart robotic exoskeleton (*SREx*), a 7 degrees of freedom exoskeleton robot designed for use by adults in upper limb rehabilitation therapy. This study implemented a model-free PID controller for *SREx*’s complicated 7 DoF. The robustness and trajectory tracking of the performance of the PID controller was evaluated through rigorous experiments with five healthy participants simulating various therapeutic scenarios of actual patients with upper arm impairment, including individual and multi-joint movement exercises with varying velocities ranging from 15 to 72°/s. Experiments were carried out in three distinct conditions: (i) The subject did not make any voluntary movements; (ii) the subject was urged to resist the motion to represent spasticity; and (iii) the subject was requested to exert abrupt jerk/perturbation to simulate involuntary movements. The statistical comparison of experimental outcomes demonstrates the efficacy of PID in robot-assisted rehabilitation. The experiment results confirm that a model-free PID controller may be employed effectively for robot-assisted rehabilitation of real patients by manipulating a therapeutic exoskeleton robot.

## Figures and Tables

**Figure 1 sensors-22-03747-f001:**
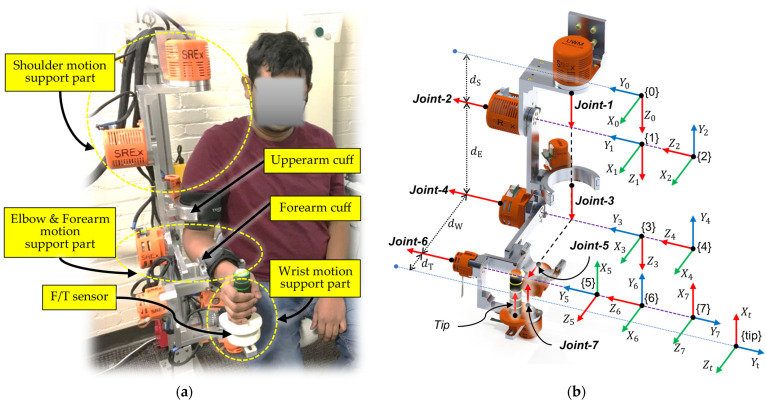
(**a**) Smart robotic exoskeleton (*SREx*); (**b**) *SREx*’s link-frame assignment, which represents the axes of rotation.

**Figure 2 sensors-22-03747-f002:**
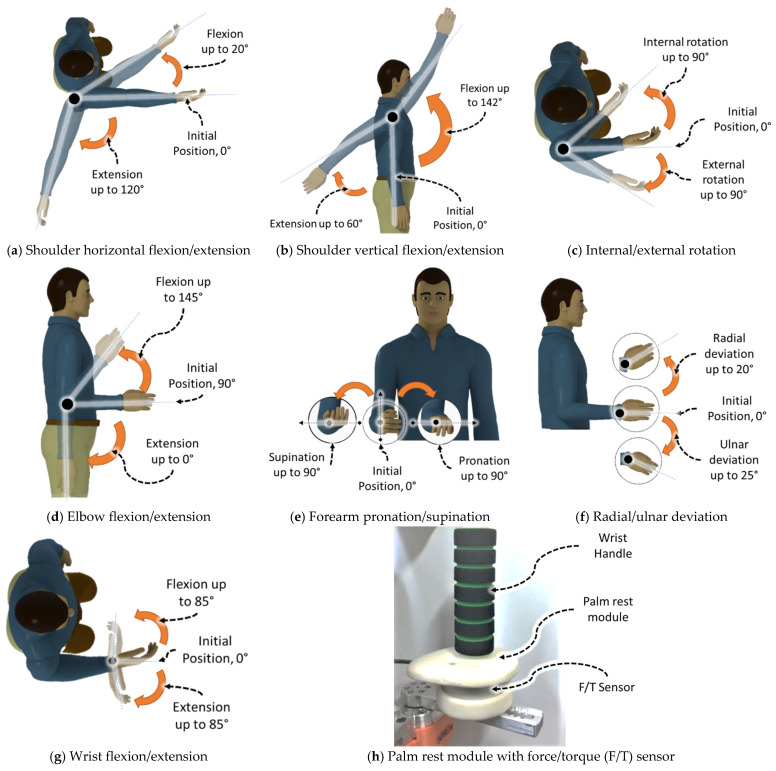
Workspace of *SREx* and associated human motions (**a**–**g**), (**h**) installed F/T sensor.

**Figure 3 sensors-22-03747-f003:**
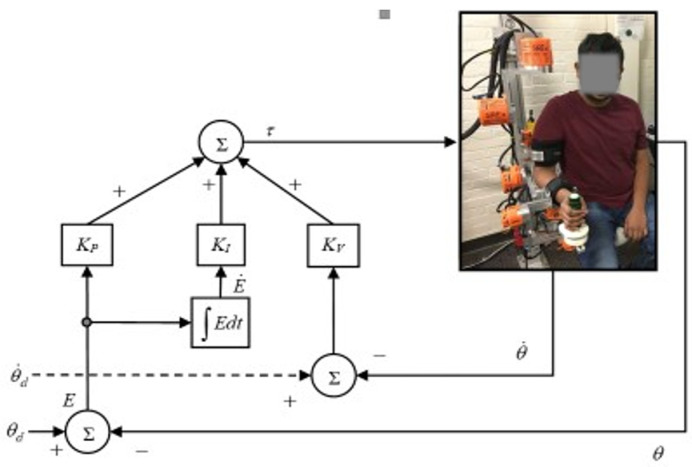
Schematic diagram of PID control.

**Figure 4 sensors-22-03747-f004:**
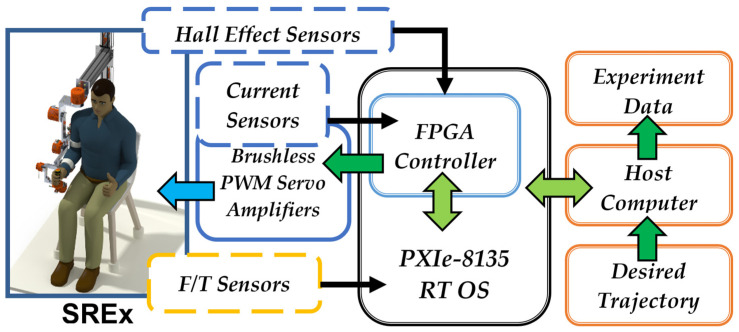
Experimental setup with *SREx* system.

**Figure 5 sensors-22-03747-f005:**
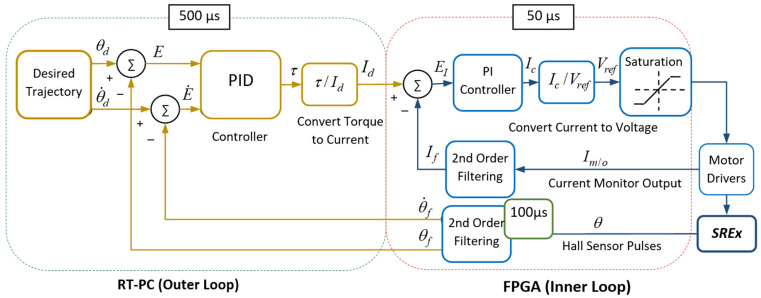
Control architecture.

**Figure 6 sensors-22-03747-f006:**
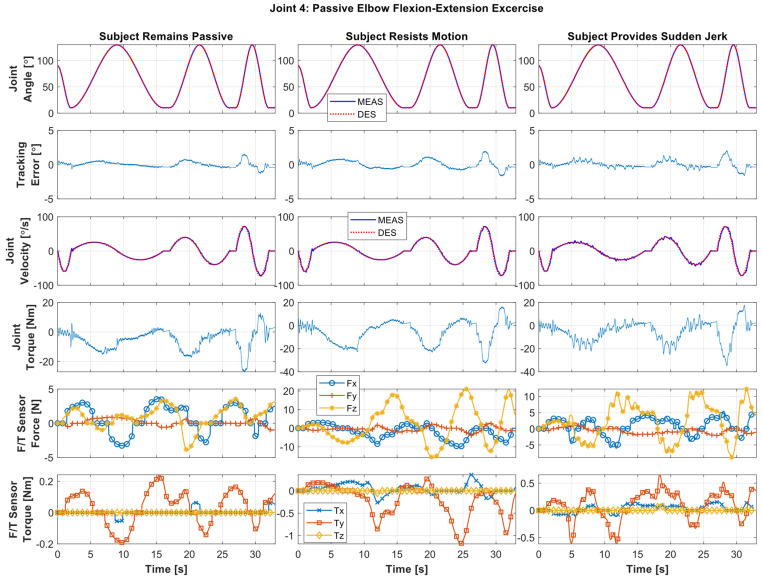
Passive elbow flexion (130°)/extension (10°) exercise for Subject A, where the subject exerts no force during the exercise (**left** column), resists the motion (**middle** column), and provides sudden jerk motion throughout the exercise (**right** column).

**Figure 7 sensors-22-03747-f007:**
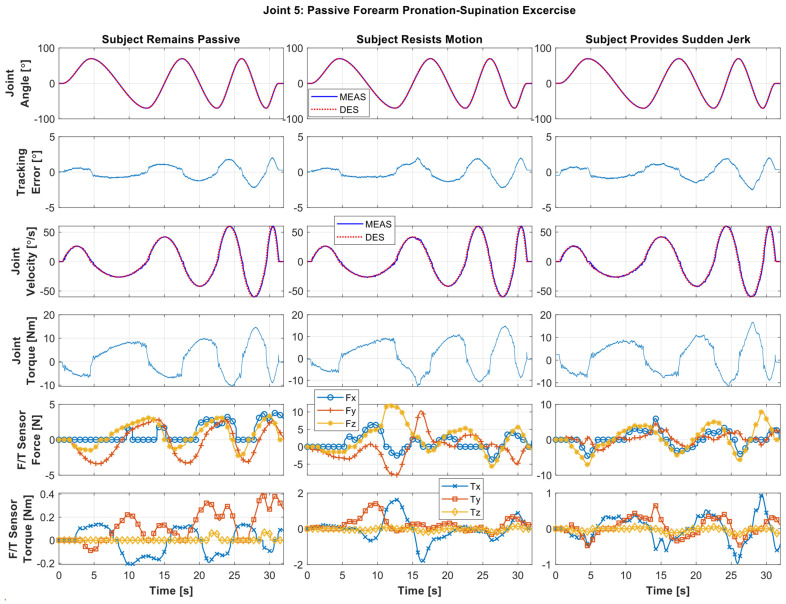
Passive forearm pronation (70°)/supination (70°) exercise where Subject C exerts no force during the exercise (**left** column), resists the motion (**middle** column), and provides sudden jerk motion throughout the exercise (**right** column).

**Figure 8 sensors-22-03747-f008:**
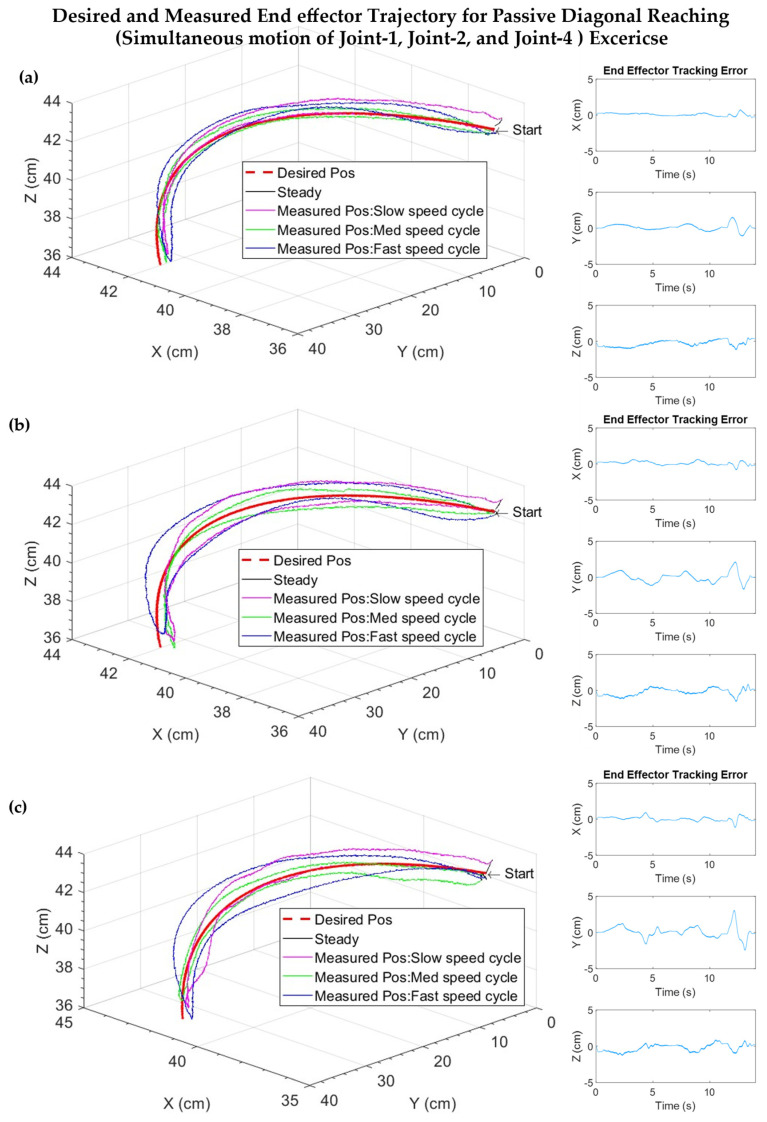
Cartesian trajectories and errors for diagonal reaching exercise with Subject B, while the subject (**a**) remains passive during the motion; (**b**) subject resists the motion; (**c**) subject provides sudden jerk force during the motion.

**Figure 9 sensors-22-03747-f009:**
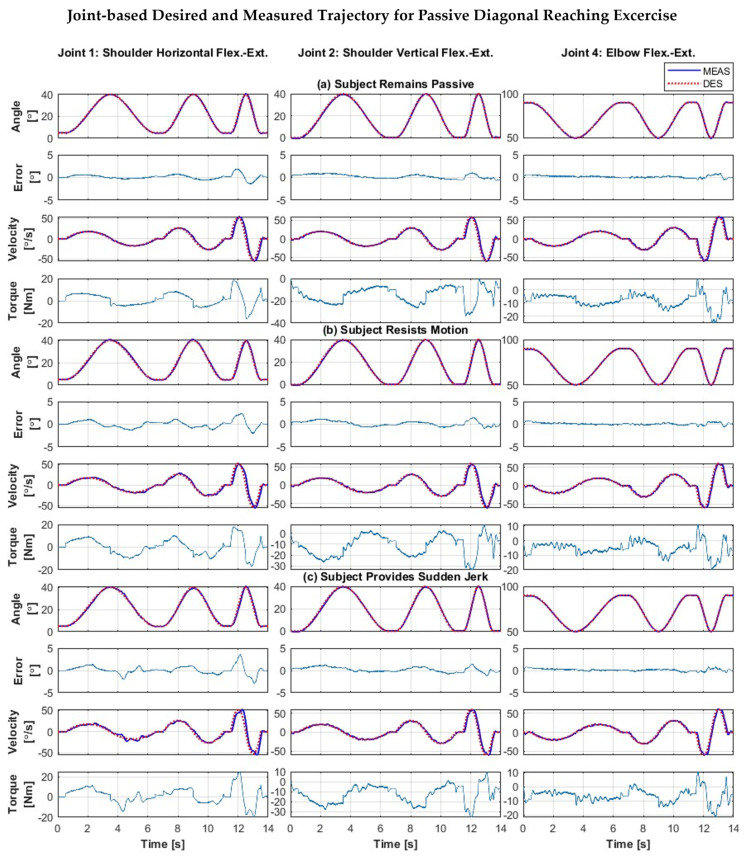
Trajectory tracking during diagonal reaching exercise (combination of shoulder vertical, horizontal, and elbow flexion/extension) of Subject B, while the subject (**a**) remains passive during the exercise motion, (**b**) resists the motion, and (**c**) provides sudden jerk force during the motion.

**Figure 10 sensors-22-03747-f010:**
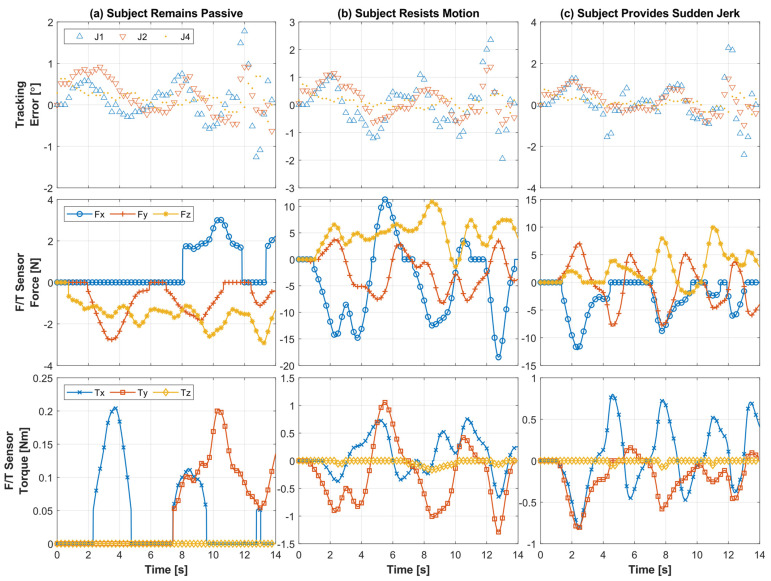
Tracking error and associated force/torque sensor data during diagonal reaching exercise of Subject B, while the subject (**a**) remains passive during the exercise motion, (**b**) resists the motion, and (**c**) provides sudden jerk force during the motion.

**Figure 11 sensors-22-03747-f011:**
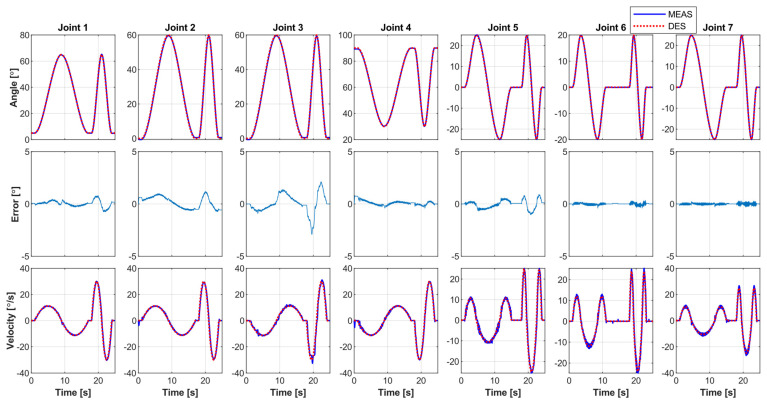
All seven joints’ passive exercise, while Subject D remains passive during the motion.

**Figure 12 sensors-22-03747-f012:**
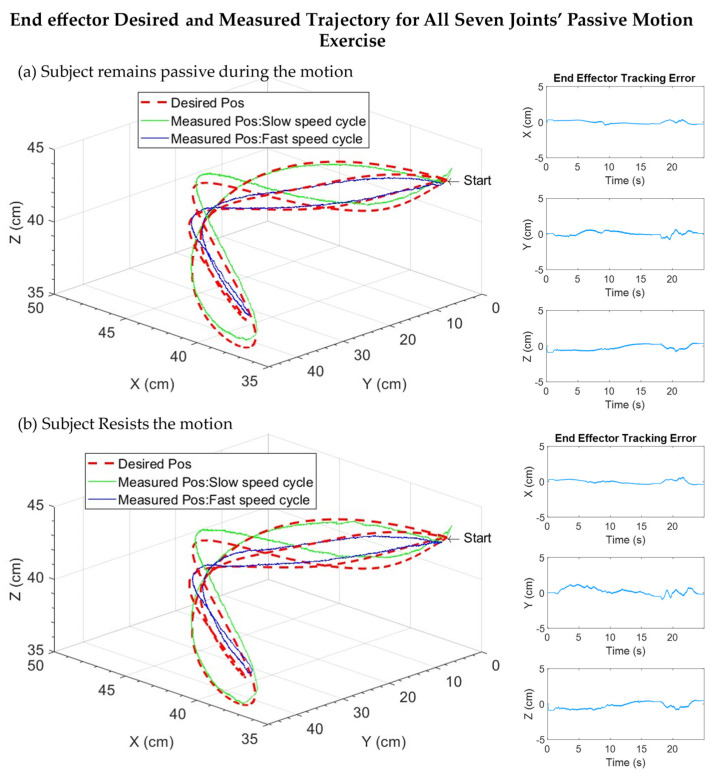
Cartesian trajectories and errors for all seven joints’ passive exercise for Subject D.

**Figure 13 sensors-22-03747-f013:**
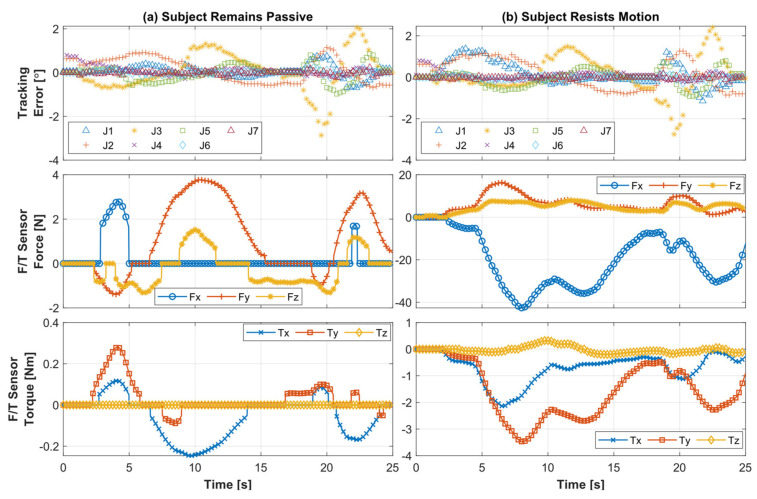
Force–torque interaction and associated tracking error for all seven joints’ passive exercise for Subject D, while the subject (**a**) remains passive during the motion and (**b**) resists the motion.

**Table 1 sensors-22-03747-t001:** *SREx*’s hardware specifications.

Actuators (Brushless), maxon precision motors, Inc., Taunton, USA				
**Specification**	**EC 90 Flat 160 W** **(Joints 1 and 2)**	**EC 45 Flat 70 W** **(Joint 3)**	**EC 90, Flat 90 W** **(Joint 4)**	**EC 45, Flat 30 W** **(Joints 5, 6, and 7)**
Nominal Voltage (V)	24	24	24	24
Nominal Speed (rpm)	2720	4860	2590	2940
Torque Constant (mNm/A)	71.2	36.9	70.5	51
Weight (g)	630	141	600	75
Strain wave Gear, Harmonic Drive LLC, Beverly, USA				
**Specification: CSF**	**17-100-2UH-LW (Joints 1 and 2)**		**14-100-2XH-F (Joint 4)**	**11-100-2XH-F (Joints 3 and 5–7)**
Torque at 2000 rpm (Nm)	24		7.8	5
Momentary Peak Torque (Nm)	108		54	25
Repeated Peak Torque (Nm)	54		28	11
Gear Ratio	100		100	100
Anti-Backlash Gear and Ring Gear (Pressure angle: 20°, Pitch: 32), Sterling Instrument, Hicksville, USA				
**Specification**	**Anti-Backlash** **Spur Gear, S1A86A-C032A062** **(Joints 3 and 5)**		**Ring Spur Gear (custom)**	
			**Joint 3**	**Joint 5**
Number of teeth	62		186	143
Bore Diameter (mm)	6.35		120	85
F/T Sensor: RFT60-HA01 with EtherCAT Adapter: RFTEC-02, Robotus, Seongnam-si, South Korea				
Parameters	Fx, Fy		Fz	Tx, Ty, Tz
Load Capacity	150 N		200 N	4 Nm
Resolution	100 mN		150 mN	5 mNm

**Table 2 sensors-22-03747-t002:** Mass and inertia properties of the *SREx*.

Link	Joints	Link Mass (kg)	Center of Gravity, CG (m)			Moment of Inertia, I (kg·m^2^)			Segment Length (m)	
			CG_x_	CG_y_	CG_z_	I_xx_	I_yy_	I_zz_		
1	1–2	2.9	0.0003	0.1290	−0.0428	0.0208	0.0136	0.0111	Base frame to Shoulder joint	0.146
2	2–3	1.2	−0.0356	−0.1419	0.0772	0.0056	0.0033	0.0065		
3	3–4	1.9	−0.0040	0.1111	−0.0213	0.0091	0.0060	0.0071	Shoulder joint to Elbow joint	0.281 ± 0.033
4	4–5	0.9	−0.0443	−0.1319	0.0417	0.0037	0.0029	0.0039		
5	5–6	0.8	−0.0074	0.0942	−0.0362	0.0057	0.0018	0.0046	Elbow joint to Wrist Joint	0.281 ± 0.035
6	6–7	0.6	0.0003	−0.1060	0.0469	0.0036	0.0024	0.0014		
7	7-Tip	0.4	0.0777	−0.0001	−0.0658	0.0007	0.0011	0.0006	Wrist joint to the tip	0.083 ± 0.041

**Table 3 sensors-22-03747-t003:** *SREx*’s workspace and modified Denavit–Hartenberg (DH) parameters.

	Joint Motion and Workspace		DH Parameters			
Joints	Motion	Joint Angles	*α_i_* _−1_	*d_i_*	*a_i_* _−1_	*θ_i_*
**1**	Shoulder joint horizontal flexion/extension	20/120°	0	*d_s_*	0	*θ* _1_
**2**	Shoulder joint vertical flexion/extension	142/60°	−π/2	0	0	*θ* _2_
**3**	Shoulder joint internal/external rotation	90/90°	π/2	*d_e_*	0	*θ* _3_
**4**	Elbow joint flexion/extension	145/0°	−π/2	0	0	*θ* _4_
**5**	Forearm joint pronation/supination	90/90°	π/2	*d_w_*	0	*θ* _5_
**6**	Wrist joint radial/ulnar deviation	20/25°	−π/2	0	0	*θ*_6_ − π/2
**7**	Wrist joint flexion/extension	85/85°	−π/2	0	0	*θ* _7_
**Tip**	Wrist handle with Force/Torque Sensor	N/A	0	0	*a_tip_*	*N/A*

where *α_i−_*_1_ is the link twist, *a_i_*_−1_ corresponds to link length, *d_i_* stands for link offset, and *θ_i_* is the joint angle of the *SREx*. Link length (*a_i_*): The length measured along *X_i_*, from axis *Z_i_* to axis *Z_i_*_+1_; link twist (*α_i_*): The angle measured along *X_i_*, from axis *Z_i_* to axis *Z_i_*_+1_; link offset (*d_i_*): The distance measured along *Z_i_*, from *X_i_*_−1_ to *X_i_*; and joint angle (*θ_i_*): The angle measured along *Z_i_*, from *X_i_*_−1_ to *X_i_*.

**Table 4 sensors-22-03747-t004:** Healthy human subject parameters.

	Age (years)	Height (cm)	Weight (kg)
Subject A	25	172.7	80
Subject B	28	177.8	82
Subject C	26	175.2	78
Subject D	44	161.5	63
Subject E	30	162.5	68

**Table 5 sensors-22-03747-t005:** Joint 4: Elbow flexion/extension passive exercise.

Mode of Exercise		Subject A *				Subject B				Subject C			
	Sensor: Force/Torque, F/T (N/Nm)	RMS F/T	Peak F/T	RMS Error *e*_4_ (°)	Peak Error *e*_4_ (°)	RMS F/T	Peak F/T	RMS Error *e*_4_ (°)	Peak Error *e*_4_ (°)	RMS F/T	Peak F/T	RMS Error *e*_4_ (°)	Peak Error *e*_4_ (°)
Passive	*F_x_*	2.08	3.76	0.42	1.49	2.86	6.75	0.38	1.49	3.16	7.67	0.43	1.60
	*F_y_*	0.46	1.01			0.49	1.23			0.98	2.01		
	*F_z_*	1.83	3.95			1.58	3.16			1.19	3.78		
	*T_x_*	0.02	0.06			0.07	0.18			0.08	0.16		
	*T_y_*	0.11	0.24			0.20	0.52			0.28	0.75		
	*T_z_*	0	0			0	0			0.01	0.06		
Resistance	*F_x_*	4.21	10.16	0.64	1.95	6.97	20.41	0.51	2.06	4.09	11.86	0.50	1.72
	*F_y_*	1.34	3.91			3.40	10.37			1.70	5.53		
	*F_z_*	9.94	21.03			8.63	24.38			3.85	10.75		
	*T_x_*	0.12	0.36			0.32	1.17			0.14	0.54		
	*T_y_*	0.42	1.19			0.60	2.47			0.37	1.24		
	*T_z_*	0.01	0.05			0.05	0.18			0.02	0.07		
Sudden Jerk	*F_x_*	2.77	6.01	0.55	2.06	5.15	16.72	0.41	1.60	3.47	9.38	0.47	1.78
	*F_y_*	1.05	2.09			2.77	9.74			1.09	2.01		
	*F_z_*	5.96	12.27			4.94	14.80			2.62	9.41		
	*T_x_*	0.08	0.16			0.28	1.16			0.08	0.22		
	*T_y_*	0.28	0.66			0.36	0.85			0.32	0.97		
	*T_z_*	0.02	0.11			0.03	0.12			0.02	0.07		

* The subject’s corresponding experimental data plots has been shown in Figure 6.

**Table 6 sensors-22-03747-t006:** Joint 5: Forearm pronation/supination passive exercise.

Mode of Exercise		Subject A				Subject B				Subject C *			
	Sensor: Force/Torque, F/T (N/Nm)	RMS F/T	Peak F/T	RMS Error *e*_5_ (°)	Peak Error *e*_5_ (°)	RMS F/T	Peak F/T	RMS Error *e*_5_ (°)	Peak Error *e*_5_ (°)	RMS F/T	Peak F/T	RMS Error *e*_5_ (°)	Peak Error *e*_5_ (°)
Passive	*F_x_*	0.25	1.77	0.87	2.06	0.38	1.98	0.83	2.06	1.67	3.8	0.94	2.18
	*F_y_*	1.55	2.77			1.46	2.77			2.12	3.42		
	*F_z_*	1.17	2.39			1.10	2.28			1.8	3.37		
	*T_x_*	0.09	0.15			0.12	0.29			0.11	0.21		
	*T_y_*	0.05	0.13			0.09	0.25			0.18	0.41		
	*T_z_*	0	0			0	0			0.02	0.07		
Resistance	*F_x_*	7.79	17.13	1.05	2.35	9.46	19.55	1.07	2.69	2.40	6.46	1.01	2.23
	*F_y_*	5.15	13.39			5.39	15.03			3.48	10.37		
	*F_z_*	3.22	7.02			6.74	16.77			4.63	12.19		
	*T_x_*	1	2.44			1.24	3.77			0.65	1.85		
	*T_y_*	0.56	1.24			0.65	1.45			0.45	1.41		
	*T_z_*	0.14	0.37			0.13	0.36			0.08	0.2		
Sudden Jerk	*F_x_*	3.2	14.68	0.94	2.58	3.93	12.06	0.95	2.64	2.21	5.99	1.00	2.52
	*F_y_*	2.84	13.90			2.96	8.44			1.5	4.51		
	*F_z_*	2.39	6.19			4.38	14.11			3.43	8.13		
	*T_x_*	0.48	1.78			0.82	2.41			0.37	0.97		
	*T_y_*	0.23	0.86			0.30	0.94			0.26	0.66		
	*T_z_*	0.04	0.15			0.05	0.21			0.06	0.19		

* The subject’s corresponding experimental data plots has been shown in Figure 7.

**Table 7 sensors-22-03747-t007:** Diagonal reaching passive exercise (composite motion of Joints 1, 2, and 4).

Mode of Exercise		Subject A				Subject B *				Subject C			
	Sensor: Force/Torque, F/T (N/Nm)	RMS F/T	Peak F/T	Mean RMS Error *e*_124_ (°)	Mean Peak Error *e*_124_ (°)	RMS F/T	Mean Peak F/T	RMS Error *e*_124_ (°)	Mean Peak Error *e*_124_ (°)	RMS F/T	Peak F/T	Mean RMS Error *e*_124_(°)	Mean Peak Error *e*_124_ (°)
Passive	*F_x_*	1.52	3.42	0.47	1.30	1.20	3.04	0.44	1.26	1.48	3.30	0.52	1.38
	*F_y_*	0.89	1.36			1.16	2.77			0.95	1.75		
	*F_z_*	0.88	1.63			1.64	2.94			1.34	2.39		
	*T_x_*	0.09	0.14			0.07	0.21			0.06	0.11		
	*T_y_*	0.14	0.31			0.08	0.20			0.15	0.28		
	*T_z_*	0	0			0	0			0	0		
Resistance	*F_x_*	5.03	10.78	0.63	1.49	8.49	18.45	0.55	1.53	2.37	6.15	0.61	1.45
	*F_y_*	3.14	5.62			4.11	8.34			1.99	4.56		
	*F_z_*	1.03	1.87			5.52	10.86			2.11	5.18		
	*T_x_*	0.31	0.55			0.36	0.75			0.22	0.52		
	*T_y_*	0.50	1.13			0.61	1.29			0.19	0.49		
	*T_z_*	0	0			0.05	0.16			0	0		
Sudden Jerk	*F_x_*	5.95	11.60	0.65	1.62	4.19	12.18	0.59	1.91	4.19	10.98	0.60	1.57
	*F_y_*	3.04	6.45			3.83	7.93			1.69	4.44		
	*F_z_*	1.05	2.37			3.65	10.10			2.02	6.10		
	*T_x_*	0.30	0.59			0.39	0.81			0.20	0.55		
	*T_y_*	0.61	1.30			0.30	0.83			0.42	1.08		
	*T_z_*	0	0			0.02	0.07			0	0		

* The subject’s corresponding experimental data plots has been shown in Figure 8, Figure 9 and Figure 10.

**Table 8 sensors-22-03747-t008:** Statistical analysis of tracking error and peak velocities, while Subjects D and E remain passive during all seven joints’ motion.

Joints	Slow Cycle (~1 to ~17 s)				Fast Cycle (~18 to ~24 s)			
	Subject D *		Subject E		Subject D *		Subject E	
	RMS Error *e_i_* (°)	Peak Velocity (°/s)	RMS Error *e_i_* (°)	Peak Velocity (°/s)	RMS Error *e_i_* (°)	Peak Velocity (°/s)	RMS Error *e_i_* (°)	Peak Velocity (°/s)
1	0.1660	11.4	0.2323	11.4	0.4776	30.3	0.5244	30.3
2	0.5538	11.3	0.5484	11.3	0.6143	29.4	0.6031	29.6
3	0.5957	12.1	0.5201	11.9	1.4149	32.5	1.4130	31.9
4	0.2710	11.5	0.2545	11.5	0.1365	30.0	0.1572	29.9
5	0.2908	11.2	0.2772	11.6	0.5898	25.6	0.5465	25.6
6	0.0616	13.1	0.0616	12.9	0.1107	25.4	0.1180	25.4
7	0.0521	11.8	0.0541	11.9	0.0947	26.7	0.1035	27.3

* The subject’s corresponding experimental data plots has been shown in Figure 11, Figure 12a and Figure 13a.

**Table 9 sensors-22-03747-t009:** Statistical analysis of tracking error and peak velocities, while Subjects D and E resist all seven joints’ motion.

Joints	Slow Cycle (~1 to ~17 s)				Fast Cycle (~18 to ~24 s)			
	Subject D *		Subject E		Subject D *		Subject E	
	RMS Error *e_i_* (°)	Peak Velocity (°/s)	RMS Error *e_i_* (°)	Peak Velocity (°/s)	RMS Error *e_i_* (°)	Peak Velocity (°/s)	RMS Error *e_i_* (°)	Peak Velocity (°/s)
1	0.5647	11.8	0.2346	11.5	0.6027	31.2	0.3703	30.5
2	0.7123	11.4	0.5742	11.4	0.7475	29.7	0.7053	29.5
3	0.5995	11.7	0.5823	12.4	1.4478	31.9	1.5938	30.2
4	0.2583	11.6	0.2772	11.6	0.1598	29.9	0.2278	29.8
5	0.3195	11.6	0.3393	11.5	0.5807	25.5	0.7222	25.6
6	0.0679	12.9	0.0786	13.0	0.1044	25.4	0.1819	25.4
7	0.0487	11.7	0.0782	12.0	0.0986	27.1	0.1659	27.6

* The subject’s corresponding experimental data plots has been shown in Figure 12b and Figure 13b.

## Data Availability

The data presented in this study are available on request from the corresponding author. The data are not publicly available due to the funded project’s scope of deliverables.

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
