# Peer review of "Robustness and Tracking Performance Evaluation of PID Motion Control of 7 DoF Anthropomorphic Exoskeleton Robot Assisted Upper Limb Rehabilitation"

_sensors, 2022, doi:10.3390/s22103747_

Round 1

Reviewer 1 Report

English changes required 

Author Response

The authors express thanks for pointing out the language issue. The authors have revised the manuscript, amended the grammatical mistakes, and rephrased the sentences for achieving better clarity and coherence of the manuscript.

Reviewer 2 Report

To provide a wide variety of upper-limb exercises to the different subjects, the author developed a 7 degrees of freedom (DoF) exoskeleton robot and implemented a model-free proportional-integral-derivative (PID) controller. There are some contents and questions about this paper as follows.

  1. It's mentioned that’ which is less than 1 o in the 1 st and 2 nd cycles, and less than 2.5 o in the 3 rd cycle’ in the 5.2.1. Individual Joint Movement Exercise(P11). What is the 1 st, 2 nd and 3 rd cycle? what is the difference between them?
  2. As a hot research area of rehabilitation robots, there are many studies on the control robustness of upper limb rehabilitation robots. Comparing the errors of other research results with the author's can better reflect the strong robustness of the control algorithm proposed by the author. At least, the author's experiment (a) (remained passive) should be compared with other studies.
  3. The paper has a lot of repetitive work or simple pictures, properly organized and reduced will be better. For example, the first pictures in Figure 6-8 can be placed on the same picture to make it clearer.

Author Response

Comment-1: It's mentioned that’ which is less than 1 o in the 1 st and 2 nd cycles, and less than 2.5 o in the 3 rd cycle’ in the 5.2.1. Individual Joint Movement Exercise(P11). What is the 1 st, 2 nd and 3 rd cycle? what is the difference between them?

Authors' response: The authors express thanks for pointing out the issue. For increasing clarity, the authors have explained the different velocity cycles in the manuscript. Please see lines #277-281, 314-318, 349-352, and 420-423. A snippet from Line #277-279 is given below, describing those cycles.

Comment-2: As a hot research area of rehabilitation robots, there are many studies on the control robustness of upper limb rehabilitation robots. Comparing the errors of other research results with the author's can better reflect the strong robustness of the control algorithm proposed by the author. At least, the author's experiment (a) (remained passive) should be compared with other studies.

Authors' response: The authors express thanks for the suggestion. The authors have compared the passive exercise results of another research for similar experiments with a 7 DoF exoskeleton type robot with this study. It was found that the conventional PID used in this research had better tracking performance than other complex model-based controllers. Please see lines #568-584.

Comment-3: The paper has a lot of repetitive work or simple pictures, properly organized and reduced will be better. For example, the first pictures in Figures 6-8 can be placed on the same picture to make it clearer.

Authors' response: The authors express thanks for the suggestion. The manuscript has been thoroughly re-organized by merging the corresponding figures. Total figure numbers have been reduced to eight from nineteen. Please see figures 6-13 (previously 6-25).

Comment-4: English improvement:

Authors' response: The authors express thanks for pointing out the language issue. The authors have revised the manuscript, amended the grammatical mistakes, and rephrased the sentences to achieve better clarity and coherence of the manuscript.

Reviewer 3 Report

First of all, I would like to mention that the work is interesting, but I have a few remarks:

  • According to Figure 4, there is no feedback from the torque generated at the joints in the control system presented in this paper. Is the control system protected in any way against overloading the operator?
  • The operating frequency of the FPGA loop is 2 kHz and the operating frequency of the controller on the RT-PC is only 200 Hz. Has a frequency analysis been carried out on the control system including the hardware section?
  • Is there susceptibility in the drive system?
  • How is the torque value measured on the individual joints?
  • The paper presents a kinematic model. Whether dynamics model also implemented in the control system?
  • Please include a diagram of the drive in your paper.
  • How a change in the configuration of the device is taken into account for the operation of the PID controller?
  • What type of filtering is used in the control circuit and what phase shift it brings to the control circuit over the operating frequency range of the device.
  • Charts 6-11 and 13-20 are of poor quality. Also, some graphs have a box, and some do not, e.g. in figure 11 torque and error have a box and other values do not. Please unify the graphic format.
  • During the complex movement shown in Figure 12, was the movement carried out by all 7 joints? in Figure 13 there is no data for the movement of joints 3,5,6,7.
  • Perhaps it would be useful to present the data from diagrams 13,15, 17 so that we can compare the work of the joints during movement: passive resistance sudden jerk

Author Response

Comment-1: According to Figure 4, there is no feedback from the torque generated at the joints in the control system presented in this paper. Is the control system protected in any way against overloading the operator?

Author response:   The authors express thanks for pointing out the issue. Figure-4 has been updated, showing the current feedback from the motor drivers. To ensure users' safety, low-level current loop output is saturated at a value such that the drivers' output does not exceed the threshold. The safety features of the SREx system have been explained in lines #128-137. In addition,  the ZB12A8 Series PWM servo amplifiers are designed to drive brushless DC motors at a high switching frequency. They are fully protected against over-voltage, over-current, over-heating, and short-circuit. Furthermore, the authors' would like to reiterate that in the 'passive' therapeutic mode which was presented in the paper, the robot follows the desired trajectory (exercise) to provide desired rehab therapy to the patient; the patient, in that case, remains passive and the robot doesn't overload the subject. In summary, the control system and the operator are protected in several checkpoints from overloaded and vice versa. 

Comment-2: The operating frequency of the FPGA loop is 2 kHz, and the operating frequency of the controller on the RT-PC is only 200 Hz. Has a frequency analysis been carried out on the control system, including the hardware section?

Authors’ response: Yes, we did. In our control architecture, we implemented the high-level control in RT-PC and low-level control in FPGA. In this case, the low-level control loop (inner loop) should be faster than the high-level control loop (outer loop). The secondary process (low-level control) must react to the secondary controller's efforts at least three or four times faster than the primary process (high-level control) reacts to the primary controller. This allows the secondary controller (low-level control) enough time to compensate for inner loop disturbances before affecting the primary process. See Section 5.1 and Figure 5.

Comment-3: Is there susceptibility in the drive system?

Authors’ response: The drive system (ZB12A8: https://www.a-m-c.com/wp-content/uploads/documents/AMC_Datasheet_ZB12A8.pdf) is quite robust. Further, to minimize/remove EMI, necessary measures were taken as standard (see this ref: https://doc.ingeniamc.com/wiki/motion-wiki/electromagnetic-interference-issues-with-servo-drive-systems). Further analog (current signals), digital (hall-effect sensor) signals are filtered before sending it to the current controller (PI), and robot controller (PID). See Section 5.1 and Figure 5.

Comment-4: How is the torque value measured on the individual joints?

Authors’ response: The individual joint torques of the robot were calculated by using the following formula:

Joint Torque = (Current sensor data) / [1000  / (Gear Ratio * Torque Constant)]

  • The current sensor data is obtained from the motor driver
  • The gear ratio is computed based on our design
  • Torque constant is available from the motor manufacturer's datasheet.

Comment-5: The paper presents a kinematic model. Whether dynamics model also implemented in the control system?

Author response:  The dynamic model implementation is out of the scope of this paper, as the control implementation presented in this paper was done with a model-free PID controller. We would like to stretch further; in this research, our goal was to use a model-free controller to provide therapies with a rehabilitation robot; obtaining accurate dynamics requires tremendous resources, and approximate or simplified dynamic models for such rehabilitation robots often do not produce the intended performance. Please see lines #60-67.  

Comment-6: Please include a diagram of the drive in your paper.

Author response: The diagram of the drive is readily available at [https://www.a-m-c.com/wp-content/uploads/documents/AMC_Datasheet_ZB12A8.pdf]. The authors have included the block diagram in the appendix section as per the reviewer's suggestion. Please see lines #218-219 and Appendix-A.

Comment-7: How a change in the configuration of the device is taken into account for the operation of the PID controller?

Author response: As we know, the PID is a model-free controller; hence, it does not require considering the model configuration of the robot/plant. The rigorous study conducted in this paper guarantees that the PID controller (with the same gains) is robust enough in maneuvering a complex exoskeleton robot (SREx) to provide different upper limb movement therapy with good tracking performance even under varying velocities and extreme disturbances.

Comment-8:  What type of filtering is used in the control circuit and what phase shift it brings to the control circuit over the operating frequency range of the device.

Author response: In the SREx system, we have used a second-order filter system; this filter is stable and easy to implement. In addition, its time of response can be manageable only by three parameters (gain, damping, and frequency). We tuned the current loop PI controller gains experimentally to achieve optimum performance. Please see lines #211-215.   See section 5.1, and  Figure -5.

Comment-9: Charts 6-11 and 13-20 are of poor quality. Also, some graphs have a box, and some do not, e.g. in figure 11 torque and error have a box and other values do not. Please unify the graphic format.

Author response: The authors express thanks for the suggestion. The manuscript has been thoroughly re-organized by merging the corresponding figures. Total figure numbers have been reduced to eight from nineteen. Please see figures 6-13 (previously 6-25).

Comment-10: 10. During the complex movement shown in Figure 12, was the movement carried out by all 7 joints? in Figure 13 there is no data for the movement of joints 3,5,6,7.

Author response: Only Joint-1, Joint-2, and Joint-4 were used for the diagonal reaching experiment. The other joints (3, 5, 6, and 7) remained steady for that particular exercise. The authors have revised subsection 5.2.2 to increase the clarity of the manuscript. Please see lines #314-410.

Comment-11: 11. Perhaps it would be useful to present the data from diagrams 13,15, 17 so that we can compare the work of the joints during movement: passive resistance sudden jerk

Author response: In the manuscript, Table 7 contains statistical data corresponding to Figures 8, 9, and 10 (previously Figure 13, 14, 15, 16, 17, and 18). The tables have been provided with relevant figure numbers and references to increase the manuscript's clarity. Please see tables 5-9.

The authors express thanks for pointing out the language issue. The authors have revised the manuscript, amended the grammatical mistakes, and rephrased the sentences for achieving better clarity and coherence of the manuscript.

Reviewer 4 Report

The paper addresses a topical issue, namely the control of the movement of an anthropomorphic exoskeleton with 7 degrees of freedom, for the rehabilitation of the human upper limb.

The title is suggestive.

The abstract is well done, the content of the paper is explained.

The strong part of the paper is that the robustness and trajectory tracking performance of the is verified by experiment

PID controller.

The introduction section is well done, with recent references in the field.

The main contribution of the paper is the introduction of a new therapeutic robot, SREx, and its control implementation with a model-independent (PID) controller to provide every variety of upper limb rehab exercises.

Section 2 describes the SREx Intelligent Exoskeleton. The workspace shown in Fig. 2 is eloquent. Also listed in Table 1 are the hardware components of the robot, the mass properties of the components in Table 2.

Section 3 considers the kinematics of the SREx robot, developed based on the Denavit-Hartenberg model. Table 3 shows SREx’s Workspace and Modified Denavit-Hartenberg (DH) parameters.

Section 4 addresses the issue of robot motion control.

Section 5 is the largest volume, as a presentation, and deals with experiments and results obtained.

The experimental setup part is shown in Fig.4.

The experimental data obtained involved a large volume of work (Fig. 6-10). These figures are explained and commented later. The analysis part for trajectory traking follows.

An analysis for the situation when we have all joints simultaneous motion is presented, followed by a statistical analysis of the data.

The paper is well written and the results obtained are conclusive. Congratulations to the authors for their research.

Author Response

The authors express thanks for pointing out the language issue. The authors have revised the manuscript, amended the grammatical mistakes, and rephrased the sentences for better clarity and coherence of the manuscript.